# Ocean Forecasting at the Regional Scale: Actual Status

Marina Tonani[1], Eric Chassignet[2], Mauro Cirano[3], Yasumasa Miyazawa[4], Begoña Pérez Gómez[5]

[1]Mercator Ocean International, Toulouse, France
[2]Center for Ocean-Atmospheric Prediction Studies, Florida State University, United States
[3]Department of Meteorology, Institute of Geosciences, Federal University of Rio de Janeiro (UFRJ), Brazil
[4]Japan Agency for Marine-Earth Science and Technology, Kanagawa, Japan
[5]Puertos del Estado, Madrid, Spain

*Correspondence to*: Marina Tonani (mtonani@mercator-ocean.fr)

**Abstract.** Operational ocean forecasting systems provide important information on physical and biogeochemical variables across global, regional, and coastal scales. Regional systems, with higher resolution than global models, capture small-scale processes like eddies and usually include tides, but lack detailed land-sea interactions essential for coastal areas. These models, often nested within global systems, vary in spatial resolution (1-20 km) and may include biogeochemical components. While regional systems focus on physical parameters such as sea surface height, temperature, salinity, and currents, only a few incorporate biogeochemical processes. The growing demand for biogeochemical data has prompted advancements and more systems will include this component in the coming years.

This paper provides a preliminary overview of the current status of regional forecasting systems, discussing examples as the Copernicus Marine Service from the OceanPredict, analysing the offer in terms of covered regions, resolution and ocean variables product catalogue.

**Short Summary:**

This article provides an overview of the main characteristics of ocean forecast systems covering a limited region of the ocean. Their main components are described, as well as the spatial and temporal scales they resolve. The oceanic variables that these systems are able to predict are also explained. An overview of the main forecasting systems currently in operation is also provided.

## 1 Introduction

Numerous oceanographic systems are providing data on physical and biogeochemical variables, spanning from global, regional to coastal scale. It can be challenging to precisely define the characteristics of a regional oceanographic system versus a global or coastal system, as there may be some overlap in the information they provide and the regions they cover. Regional models

typically offer greater detail than global models due to their higher resolution and ability to capture small-scale processes such as eddies, fronts, and local features. This approach avoids the significant computational costs associated with running a global system at high resolution. Additionally, most regional models incorporate tides, which are not always included in global models. Moreover, they can be optimized for specific areas, which may have unique oceanographic characteristics and require higher resolution or tailored parameterizations (Tonani et al., 2015). However, they do not include the processes of land-sea interaction that are important for the coastal areas, e.g. the dynamics of nearshore currents, sediment transport, the delta/estuary processes, and some biogeochemical processes, typically solved by coastal systems. In addition, the spatial scale is a factor in differentiating global, regional and coastal. Regional systems are directly nested into global and may or may not have nested coastal systems. In recent years, various approaches have been developed to increase model resolution only where needed, leveraging unstructured grid models. These models show great promise in balancing the need for high-resolution detail with manageable computational costs. As a result, the distinction between regional and coastal models has become less defined. However, differences in the processes resolved and key parameterizations remain essential for accurately representing coastal dynamics and processes versus regionals. Another promising development is the use of machine learning-based forecasting systems and hybrid models. Once properly trained, these systems can deliver accurate forecasts while significantly reducing computational costs. Although most of these systems are still under development or in pre-operational stages, they are expected to be integrated into the landscape of operational forecasting systems in the near future.

Several regional forecasting systems have been developed across the world and are currently in operations (Tonani et al., 2015; Schiller et al., 2015; Alvarez-Fanjul et al., 2022). A brief overview of the main characteristics of these systems is presented in Sections 2 and 3. Section 4 provides details on the regional systems described by OceanPredict (Tonani et al., 2015; Bell et al., 2015), and the Copernicus Marine Service (Le Traon et al,, 2019), considered a representative overview of the systems currently in operation. Providing an exhaustive account of all the regional forecasting systems is outside the scope of this document and would require a dedicated survey. This need is fullfilled by the Atlas initiative (https://www.unoceanprediction.org/en/atlas/), launched few months ago by the OceanPrediction Decade Collaborative Centre (OceanPrediction DCC) aiming to map all the operational forecasting centres and their characteristics.

## 2 General characteristics

There are several factors that determine the spatial scale of a regional ocean forecasting system, including the region's size, bathymetry, and oceanographic characteristics, as well as the system's purpose. Operational systems currently have resolutions ranging from approximately 1 to 20 kilometers. Usually larger regions don't need the same fine resolution as smaller regions, and so can cope with a coarser resolution. Shelf sea regional systems may require a finer spatial resolution compared to larger regions such as the North Atlantic basin. For example, in shelf areas, smaller grid cells of around 1 kilometer are necessary, whereas in the North Atlantic, larger grid cells of 10 kilometers or more are enough.

The resolution needed by a model grid for resolving the baroclinic eddy dynamics can be computed as function of the first baroclinic Rossby radius of deformation, Rd. A well establish metric used for assessing this relationship, (Hallberg et al 2013) is: Rh=Rd$\sqrt{(\Delta x^2 + \Delta y^2)/2}$ where Rd is the first baroclinic Rossby radius of deformation and $\Delta x$ and $\Delta y$ is the horizontal grid spacing of the model. A model is defined eddy-resolving when Rh > 2 , otherwise it is eddy permitting.

The choice between a regional, global, or coastal oceanographic system will depend on a variety of factors, including the specific operational needs of the user, the oceanographic characteristics of the region of interest, and the computational resources and data availability. Regional forecasting systems must be tailored to the specific processes characterizing their target areas. This requires selecting appropriate parameterizations and designing system components accordingly. In some cases, coupling additional components may be justified if the resulting improvement in forecast accuracy outweighs the associated computational costs.

Design, components, and configurations of these systems can vary widely. Most of them use an ocean general circulation model such as NEMO (Madec et al., 2022), ROMS, or HYCOM, and data assimilation components based on the Kalman filter or variational methods. Additionally, some systems include wave and biogeochemical model components. These model components can be standalone or coupled in various configurations. Most of them rely on atmospheric fields at the ocean/atmosphere boundaries because they are not coupled with an atmospheric model. Biogeochemical components are a standard feature in all the European systems of Copernicus Marine Service, but they are missing in most other systems. Some countries, such as India, are currently developing a biogeochemical component for future use.

Regional models are often nested into a global or another regional system, parent model, providing them with lateral boundary forcing. Many systems, in turn, provide lateral boundaries and initialization fields to coastal systems.

Most systems provide deterministic forecasts, although a few already have the ability to produce ensemble forecasts. There is a growing interest in developing systems that can produce ensemble forecasts.

The forecast production is daily for most systems, although some run them twice per day. The forecast lead time is typically between 5 and 10 days (short-medium range) (WMO, 2021). The time resolution of their products varies from hours to days, with some fields delivered at a higher frequency of 15 minutes.

Ultimately, the spatial and temporal scales of a regional ocean forecasting system, as well as the selection of its components, will depend on the region's specific needs and characteristics.

## 3 Oceanographic information provided by regional systems

The regional oceanographic services play a crucial role in measuring the Ocean Essential Variables (EOV) defined by the Global Ocean Observing System (GOOS). EOVs are classified into four categories: physics, biology/ecosystems, biogeochemistry, and cross-disciplinary. This description is mainly focused on short term forecasting products, because most systems do not provide long climatological series of the past to understand how ocean conditions are changing over time. Several regional reanalysis studies exist, but obtaining information about the services delivering this data can be challenging.

Copernicus Marine Services offers an operational service for reanalysis produced by all its regional systems, updated at least annually. However, additional services are also available. In this context, the Ocean Prediction DCC Atlas will be instrumental in providing detailed and structured information on these systems.While the regional forecasting systems primarily focus on physical parameters such as temperature, salinity, currents, and sea level, some also include wave and sea ice components to provide comprehensive information about the ocean's physical characteristics.

It is important to clarify that most regional systems forecast sea level, also referred to as Sea Surface Height. This represents the distance between the ocean surface and a reference mean sea level. This reference mean sea level depends, at each individual grid point, on the model domain and its physics (barotropic vs. baroclinic, consideration of tides, wind parameterization, etc), as well as on the physics and characteristics of the parent model. This should be considered when comparing model data with observations (e.g. tide gauge data usually referred to national or local datums) or other models (e.g., regional versus coastal models). Additionally, approximations made by the models and their parameterization, and data assimilation schemes can impact the accuracy of this information."Except for the Copernicus Marine Service, most regional systems do not deliver information on biogeochemistry and biology. These models are computationally very expensive due to the high number of variables and processes they take into account, preventing them from providing in most cases the level of details and accuracy that users require. However, despite these limitations, there is a growing recognition of the importance of monitoring and understanding biogeochemical variables in the ocean as confirmed by the steady increase in the demand for the biogeochemical products at Copernicus Marine Services. Additional regional systems, i.e. the Indian INDOFOS and Australia, are currently developing a biogeochemical model that will be coupled to their systems.

## 4 Operational regional systems across the world

Different countries and organizations have developed regional ocean forecasting systems. The European Copernicus Marine System (Le Traon et al., 2019), since 2015, has a set of regional systems that cover all the European seas, the Arctic ocean, and the northeastern Atlantic. Australia has a relocatable regional system for refining its global model around its own region. Other countries such as Brazil (Franz et al., 2021; Lima et al., 2013), Canada, China, India, Japan (Sakamoto et al., 2019), Korea, and the US have regional ocean forecasting systems or a set of them, covering the ocean and seas surrounding their coasts.

These systems use different data sources and modeling techniques, but they also have many similarities. Table 2.2-1 provides a non-exhaustive summary of the regional systems as described by OceanPredict and by the Copernicus Marine Service.

As described in Section 1, their geographical extension can vary from relatively small surfaces to extended areas and their horizontal grid resolution is usually of the order of 2-20 km. They do all provide the standard physical variables but only few also provide biogeochemical information.

Differences also exist in the level of operational readiness among the systems described, as well as in their product validation procedures and data dissemination policiesNot all this information has an open and free access policy but all the regional systems play an important role in monitoring and forecasting the ocean.

**Table 1: Summary of the regions covered by the regional ocean forecasting systems based on the information available from OceanPredict and from Copernicus Marine Service. The last column describes the Ocean Essential Variables (defined by GOOS) provided by each system.**

| Country/ Provider | Geographical area/System | Resolution | Essential Ocean Variables |
|---|---|---|---|
| Australia - Blue Link  | Relocatable regional model along Australian coast | ~2km | Physics (T, S, currents, SSH, waves) Biogeochemistry under development |
| Brazil – REMO  | • Atlantic ocean<br>• Brazilian continental Margin (METAREA V) | • 1/12°<br>• 1/24° | Physics (T, S, currents, SSH) |
| Canada-Concept RIOPS  | • Arctic<br>• North Atlantic and Great Lakes | • 1/4°<br>• 1/36° | Physics (T, S, currents, SSH, sea ice) |
| China – NMEFC  | • Northwest Pacific<br>• Bohai Sea, Yellow Sea and East China Sea<br>• South China Sea | • 1/20° (1/36°)<br>• 1/30°<br>• 1/30° | Physics (T, S, currents, SSH) |
| Europe – Copernicus Marine Service  | • Arctic Sea<br>• Baltic Sea<br>• North West European Shelf<br>• Iberian-Biscay-Irish sea | • 3-6 km<br>• ~2km<br>• ~2 and 7km<br>• ~2-3 km<br>• ~5-3 km<br>• ~3km | Physics (T, S, currents, SSH, sea ice, waves) Biogeochemistry (nutrients, oxygen, carbonate system, organic carbon, optics) |

| | | | |
|---|---|---|---|
| | • Mediterranean Sea<br>• Black Sea | | Biology (plankton) |
| India – INCOIS<br> | • Indian Ocean (INDOFOS)<br>• Local Indian Ocean regions (HOOFS)<br>• Indian Ocean nested into Global (ITOPS-IO) | • 1/12°<br>• 1/48°<br>• 1/16° | Physics (T, S, currents, SSH)<br>Biogeochemistry – under development |
| Japan – MOVE/MRI.COM<br> | • Japanese area<br>• North Pacific | • 1/33° x 1/50°<br>• 1/10° x 1/11° | Physics (T, S, currents, SSH) |
| Republic of Korea<br> | • North Pacific<br>• The Yellow and East China Sea (KOOFS) | • 1/28°<br>• 3 km | Physics (T, S, currents, SSH) |
| US – NOAA<br> | • West Coast Operational Forecast System (WCOFS) | 4 km | Physics (T, S, currents, SSH) |

**Competing interests**

The contact author has declared that none of the authors has any competing interests

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
