# Peer review of "Ocean Forecasting at the Regional Scale: Actual Status"

_State of the Planet, 2024_

## Author Response (AR1)

**RC1**: 'Comment on sp-2024-30', Anonymous Referee #1, 13 Oct 2024  reply

**Review of Ocean Forecasting at the Regional Scale: Actual Status**

**General Comments**

This is a review of the paper by Tonani et al. "Ocean Forecasting at the Regional Scale Actual Status". The authors have a given an overview of operational regional ocean forecasts systems from around the world. They first describe regional systems, as being between global and coastal scale systems. They have then compiled a list of systems from many different countries, and compare and contrast them in terms of resolution, complexity, number of component systems (physical 3d ocean, biogeochemistry, waves, etc) and data assimilation. They then tabulate this information.

This is potentially a very useful report, however, I would suggest providing a little more information would substantially increase its usefulness.

**Specific Comments**

I suggest they include a world map showing all the model domains. Thanks

I don't think it is clear how exhaustive this list of systems is. Do they mean to give an example list of operational regional systems? or do they mean to provide a definitive list – this would me more difficult to do, but more useful. I wondered if there were systems covering South Africa, Indonesia, New Zealand, European systems not included in Copernicus Marine, east coast US, Gulf of Mexico Russia etc. Perhaps this could be made clear in the report somehow. Showing a map of the domains may make it clear where systems may be missing.

The table of info could be expanded (or an additional table added into the appendices, showing:

- Is the data (freely) available?

    o What data?

    o The URL of the data?

- Is there Data Assimilation?

    o What data is assimilated?

- What the lead time of the forecast is?

    o How much historic data is available?

- Is it an ensemble system, how many members?

- What model is based on?

I think these points should be considered before publication.

*ANSWER: Thanks for this important remark. The OceanPrediction Decade Collaborative Centre (OceanPrediction DCC) Atlas will address this need through its digital platform, which uses a standardized format for gathering relevant information. Collecting this data independently would duplicate the Atlas's efforts and require direct engagement with numerous centers. As a digital*

**Technical Corrections**

There are also a few places where the language could be tightened up. Below are some examples.

Line 20:

"overview **on** status of regional forecasting systems **at today**, discussing examples as the Copernicus"

"overview of the current status of regional forecasting systems, discussing examples such as the Copernicus"

ANSWER: Thanks, the text has been corrected.

Line 40

"Regional systems are directly nested into global and might have or not nested coastal systems."

"Regional systems are directly nested into global and may or may not have nested coastal systems."

*ANSWER: thanks, the text has been corrected.*

Line 50:

For instance, shelf sea regional systems may require a finer resolution, with smaller grid cells around 1 kilometer, while a larger region such as the North Atlantic **may need coarser resolution** with larger grid cells around 10 kilometers or more

This sounds like you mean that larger region need a coarser resolution, and so would be less accurate etc with the same fine resolution. I think you mean that larger regions don't need the same fine resolution as smaller regions, and so can cope with a coarser resolution.

*ANSWER: Thanks, the text has been re-worded as follow: "Shelf Sea regional systems may require a finer spatial resolution compared to larger regions such as the North Atlantic basin. For example, in shelf areas, smaller grid cells of around 1 kilometer are necessary, whereas in the North Atlantic, larger grid cells of 10 kilometers or more are enough."*

Line 72: surely EOV is Essential Ocean Variable?

*ANSWER: Yes, the regional oceanographic services play a crucial role in measuring the Ocean Essential Variables (EOV) defined by the Global Ocean Observing System (GOOS).*

Line 76:

Not sure what you mean by "There are several regional reanalysis studies, but it is not easy to map the services delivering this information." Do you mean its not easy to compare regional forecasts to

regional reanalyses are they are not quite compatible? If so, I don't think map is the right word, or if it is, it needs more clarification

*ANSWER: Thanks for the comment. We have re-worded as: "Several regional reanalysis studies exist, but obtaining information about the services delivering this data can be challenging. Copernicus Marine Services offers an operational service for reanalysis produced by all its regional systems, updated at least annually. However, additional services are also available. In this context, the Ocean Prediction DCC Atlas will be instrumental in providing detailed and structured information on these systems."*

Line 82, 83. These couple of sentences need to be tightened up a bit;

*ANSWER: Thanks it has been slightly re-worded as: "It is important to clarify that most regional systems forecast sea level, also referred to as Sea Surface Height. This represents the distance between the ocean surface and a reference level, typically the geoid in these models. Different models may use different datums for sea level due to different reference level of the parent models they are nested in. This should be considered when comparing model data with observations or other models (e.g., regional versus coastal models). Additionally, approximations made by the models and data assimilation schemes can impact the accuracy of this information."*

Line 93, perhaps Additional should be other?

*ANSWER: Thanks, your comment has been taken onto consideration and additional replaced by others.*

**RC2**: 'Comment on sp-2024-30', Anonymous Referee #2, 21 Oct 2024  reply
**General Comments**

The article offers a concise overview of the capabilities of the Copernicus Marine Service and Ocean Predict's operational regional systems. It emphasizes the significance of regional modeling and operational forecasting systems for both end-users and intermediate users working with coastal and local models, services, and applications. The main features of these models are outlined, including their areas of implementation, temporal coverage, and horizontal resolution.

The article effectively underscores the importance of regional model resolution and its ability to capture small-scale processes, while also addressing the limitations of unresolved land-ocean interactions and nearshore dynamics. However, the discussion could be further enriched by exploring the relevance of other model characteristics beyond resolution.

Overall, the paper is informative and provides a valuable summary, but it would benefit from the inclusion of additional details to enhance its comprehensiveness.

**Specific Comments**

The "Introduction" section does not mention other types of modeling systems, such as unstructured grid models or machine learning/artificial intelligence (ML/AI)-based regional models, which could offer alternative approaches to achieving higher resolution in coastal areas. Including this discussion would broaden the scope of the paper.

*ANSWER: Thanks for the comment. The following text has been added to the introduction:*

*"In recent years, various approaches have been developed to increase model resolution only where needed, leveraging unstructured grid models. These models show great promise in balancing the need for high-resolution detail with manageable computational costs. As a result, the distinction between regional and coastal models has become less defined. However, differences in the processes resolved and key parameterizations remain essential for accurately representing coastal dynamics and processes versus regionals. Another promising development is the use of machine learning-based forecasting systems and hybrid models. Once properly trained, these systems can deliver accurate forecasts while significantly reducing computational costs. Although most of these systems are still under development or in pre-operational stages, they are expected to be integrated into the landscape of operational forecasting systems in the near future."*

While the authors emphasize the importance and limitations of model resolution, they do not address the key factors that differentiate between eddy-resolving and eddy-permitting models. Including this distinction would provide further clarity. This shall be included .

*ANSWER: Thanks for pointing this out. This sentence has been added:*

*"The resolution needed by a model grid for resolving the baroclinic eddy dynamics can be computed as function of the first baroclinic Rossby radius of deformation, Rd. A well establish metric used for assessing this relationship, (Hallberg et al 2013) is: Rh=Rd$\sqrt{(\Delta x^2 + \Delta y^2)/2}$ where Rd is the first baroclinic Rossby radius of deformation and $\Delta x$ and $\Delta y$ is the horizontal grid spacing of the model. A model is defined eddy-resolving when Rh > 2 , otherwise it is eddy permitting."*

In addition to focusing on model resolution, the article would benefit from a more in-depth analysis of the key features to be addressed when implementing regional ocean forecasting systems. This should include considerations such as parameterization

choices, the processes represented, model coupling, and the complexity of the model in relation to its objectives and intended applications. –

*ANSWER: Thanks, the following text has been added: "Regional forecasting systems must be tailored to the specific processes characterizing their target areas. This requires selecting appropriate parameterizations and designing system components accordingly. In some cases, coupling additional components may be justified if the resulting improvement in forecast accuracy outweighs the associated computational costs."*

When referring to lower-resolution systems (often global, but not always) in which regional systems are nested, it would be preferable to use the term "parent model" rather than "global model." This would account for cases where regional systems are nested within larger regional models rather than global ones.

*ANSWER: Yes, done the text has been modified accordingly.*

Adding more model characteristics and details to the summary table would greatly enhance its utility for the reader. Important details such as vertical resolution, model type, data assimilation schemes, assimilated data, and atmospheric forcing are briefly mentioned in the text but should be explicitly included in the table for easy reference. Adding a reference to the data availability (link to data, and/or DOI) would be also useful.

*ANSWER: Thanks for this important remark. The OceanPrediction Decade Collaborative Centre (OceanPrediction DCC) Atlas will address this need through its digital platform, which uses a standardized format for gathering relevant information. Collecting this data independently would duplicate the Atlas's efforts and require direct engagement with numerous centers. As a digital platform, the Atlas ensures that information remains up to date, well-organized, and easily accessible. We believe referencing the Atlas—maintained for long-term use—is a more efficient approach to sharing this information. We are committed to supporting and collaborating with OceanPrediction DCC to keep the Atlas updated and continuously improved, strengthening global access to ocean forecasting capabilities and products. Currently, the majority of the systems discussed in this paper are already referenced in the OceanPrediction DCC Atlas.*

A sentence has been added to the last paragraph of the Introduction, stating: "This need is fulfilled by the Atlas initiative (https://www.unoceanprediction.org/en/atlas/), launched few months ago by the OceanPrediction Decade Collaborative Centre (OceanPrediction DCC) aiming to map all the operational forecasting centers and their characteristics."

L15: please specify how you consider tides as small-scale processes. Is it referred to interaction of tides with small-scale processes?

*ANSWER: Thanks for pointing out this inaccuracy. The text has been re-wroded as follow:* "*Regional systems, with higher resolution than global models, capture small-scale processes like eddies and include tides,*"

L17: I would suggest to add also salinity among the list of physical variables

*ANSWER: Thanks, for pointing this omission. We added salinity.*

L34: please specify also here why including tides as small scale process

*ANSWER: Thanks for the correction. Tides are not small scale processes. They sentence as been re-worded as follow:* "*Regional models typically offer greater detail than global models due to their higher resolution and ability to capture small-scale processes such as eddies, fronts, and local features. This approach avoids the significant computational costs associated with running a global system at high resolution. Additionally, most regional models incorporate tides, which are not always included in global models.*"

L84: I suggest to use" different reference level" instead of "different physics"

*ANSWER: Thanks for the comment, the sentence has been re-worded as:* "*It is important to clarify that most regional systems forecast sea level, also referred to as Sea Surface Height. This represents the distance between the ocean surface and a reference mean sea level. This reference mean sea level depends, at each individual grid point, on the model domain and its physics (barotropic vs. baroclinic, consideration of tides, wind parameterization, etc), as well as on the physics and characteristics of the parent model. This should be considered when comparing model data with observations (e.g. tide gauge data usually referred to national or local datums) or other models (e.g., regional versus coastal models). Additionally, approximations made by the models and their parameterization, and data assimilation schemes can impact the accuracy of this information*"

L86 and 87 I would suggest to also add few words on the choice of different parameterizations in addition to the approximations

ANSWER: *Thanks, this comment has been taken into account, re-wording as* "*Moreover, it's worth taking into account that the approximations done by the models and data assimilation schemes and the differences in the parameterizations used by the models can have an impact on the accuracy of this information.*"

L107: I would suggest to also refer to the validation procedures.

*ANSWER: Thans, we have taken this important comment into account:* "*Differences also exist in the level of operational readiness among the systems described, as well as in their product validation procedures and data dissemination policies*"

**Technical Corrections**

L31: substitute "and" with "to" coastal scale

*ANSWER: thanks, correction taken into account.*

L110 summary of the "regions" instead of "region"

*ANSWER: thanks, correction taken into account.*